# Patients with central serous chorioretinopathy have high circulating alpha-klotho concentrations

**Eri Tahara, Yuki Yamamoto●\*, Takaaki Sugisawa, Fumi Gomi●**

Department of Ophthalmology, Hyogo College of Medicine, Nishinomiya, Hyogo, Japan

\* yuki.kom0923@gmail.com

## Abstract

Stress is a risk factor for central serous chorioretinopathy (CSC), but a suitable biomarker of this stress has not been identified. We aimed to evaluate alpha-klotho (αKl) as a potential biomarker of CSC. The circulating concentrations of αKl in patients diagnosed with acute or chronic CSC and treated at Hyogo College of Medicine between December 2019 and July 2021 were retrospectively compared with those of healthy individuals. We also compared the αKl concentrations of patients with acute or chronic CSC. Furthermore, we evaluated the relationships of age, sex, smoking status, and subfoveal choroidal thickness (SFCT) with αKl concentration. Patients in whom subretinal fluid reaccumulated in the same eye after its resolution were defined as having recurrent CSC. We studied 56 patients (46 men and 10 women) and 27 healthy controls (19 men and 8 women); and 38 and 18 eyes with acute and chronic CSC, respectively. The mean circulating concentration of αKl was higher in patients with CSC than in controls (827±232 and 724±183 pg/mL, respectively; $p = 0.035$). The mean SFCT was greater in patients with CSC than in controls (416±91.0 and 278±96.3 μm, respectively; $p<0.0001$). The mean αKl concentration was significantly higher in the patients with acute CSC than in those with chronic CSC (898±221 and 740±224 pg/mL, respectively; $p = 0.038$). Recurrence of CSC occurred in 10 of 56 (17.9%) eyes, of which five eyes were in the acute CSC group and five were in the chronic CSC group. Patients who experienced recurrence had significantly higher αKl concentrations than those who did not ($p = 0.0219$). There were no significant relationships of αKl concentration with age, sex, smoking history, or SFCT. In summary, the circulating αKl concentrations of patients with CSC are high, which suggests that αKl may be an indicator of stress in such patients.

## Introduction

Central serous chorioretinopathy (CSC) is a self-limiting retinopathy characterized by neuroepithelial detachment and dysfunction of the retinal pigment epithelium (RPE) [1]. The risk factors for CSC include a thick choroid, a type A behavior pattern, and the use of steroids [2]. However, the underlying pathophysiology of CSC is not fully understood [2, 3]. Stress is a

**Data Availability Statement:** All relevant data are within the paper and its Supporting Information files.

**Funding:** The authors received no specific funding for this work.

**Competing interests:** The authors have declared that no competing interests exist.

known risk factor for CSC [4], but no biomarkers of stress have been identified for use in such patients.

Exposure to high concentrations of endogenous or exogenous glucocorticoid is considered to be a potential cause of CSC [5]. In addition, the serotonergic system is thought to be involved in the pathophysiology and treatment of stress-related disorders, such as anxiety and depression. In addition, Nakanishi *et al.* proposed that the serum concentration of alpha-klotho (αKl) might represent an objective biomarker of stress [6].

*KL*/klotho is an aging gene that encodes αKl, which is expressed in the distal tubules of the kidneys, parathyroid gland, and choroid plexus, and has been linked to heart disease, type 2 diabetes, and cerebrovascular disease [7, 8]. The circulating αKl concentration has also been shown to increase in response to psychological stress and smoking, and this may represent a protective response to adverse events in healthy individuals [9]. Therefore, the circulating αKl concentration may be an indicator of stress. To evaluate this possibility, in the present study, we aimed to compare the αKl concentrations of patients with CSC and healthy adults.

## Materials and methods

### Study design and patient eligibility

The study conformed with the principles of the Declaration of Helsinki and we obtained approval from the Ethics Committee of Hyogo Medical University (approval number 3385) before gathering the data and conducting the analysis. We obtained written consent from the patient. We conducted a retrospective study of the medical records of patients diagnosed with acute or chronic CSC who were treated at the Hyogo College of Medicine, Japan, between December 2019 and July 2021.

The diagnosis of CSC was based on the detection of serous retinal detachment and thickening of the choroid on optical coherence tomography (OCT) imaging and idiopathic leaks at the level of the retinal pigment epithelium (RPE) on fluorescein angiography (FA) imaging. Indocyanine green angiography (ICGA) was also performed, which confirmed the choroidal vascular hyperpermeability. We included patients with both acute and chronic CSC, which we categorized based on a duration of symptoms of less or more than 6 months, respectively. Patients with macular neovascularization detected using optical coherence tomography angiography (OCTA) at baseline; those who had used steroids; those with systemic diseases, such as renal or cardiac disorders, which affect αKl concentration; those with a history of anti-vascular endothelial growth factor drug therapy; and those with severe myopia of >−6.0 D were excluded. Age-matched controls were selected from among individuals who indicated their consent and were deemed to be healthy, with no previous history of CSC or other systemic diseases, such as heart failure or renal dysfunction.

### Study protocol

The data collected from the medical records included the age, sex, steroid usage, and smoking history of both the patients and controls. The circulating concentrations of αKl had been measured in both the patients and controls. To ensure uniformity of the conditions, after a diagnosis of CSC had been made by FA and ICGA, blood samples were drawn approximately 6 hours after waking from patients who gave their consent, and at the time consent was given in controls. All the patients had undergone ophthalmic examinations, including slit-lamp biomicroscopy, FA/ICGA using a confocal laser scanning instrument (HRA2; Heidelberg Engineering, Heidelberg, Germany), OCT, and OCTA using a swept-source OCT (DRI-OCT, Topcom, Tokyo, Japan). A psychological test (the Profile of Mood States (POMS) questionnaire) was also completed by both the patients and controls. This consisted of 65 questions to assess

Anger-Hostility (AH), Confusion-Bewilderment (CB), Depression-Decline (DD), Fatigue-Apathy, Tension-Anxiety (TA), Vigor-Activity (VA), and Friendliness (F), as well as to provide a total mood disturbance (TMD) score that represents the overall negative mood state. We categorized CSC as acute or chronic, based on a duration of symptoms of less or more than 6 months, respectively. The initial treatment options were PDT (PDT alone or PDT in combination with anti-VEGF/focal laser) and focal laser photocoagulation. Patients whose subretinal fluid was improving after referral to our hospital, and those in whom spontaneous resolution could be expected, were followed up without treatment. Patients who experienced a recurrence of subretinal fluid accumulation in the same eye that had previously disappeared following the measurement of αKl concentration were defined as having recurrent CSC. The subfoveal choroidal thickness (SFCT), defined as the distance between the Bruch membrane and the choroidoscleral boundary, was measured using swept-source OCT.

## Comparisons made

We compared the αKl concentrations of the patients with CSC and the healthy controls, and those of patients with acute or chronic CSC. In addition, we evaluated the relationships of age, sex, smoking, and SFCT with αKl concentration.

## Statistical analyses

The mean and median values and the range were calculated for continuous data, and the number of values and the percentage for each category were calculated for categorical data. Differences between groups were identified using Student's $t$-test or the Wilcoxon signed-rank test for continuous data and Fisher's exact test or the Pearson $\chi^2$ test for categorical data. The logarithm of the minimum angle of resolution (log MAR) was used for statistical analysis. All the statistical analyses were performed with JMP® Pro (version 15.0.0, SAS Institute Inc., Cary, NC, USA). Statistical significance was accepted when $p < 0.05$.

## Results

A total of 56 patients (46 men and 10 women) and 27 healthy controls (19 men and 8 women) were included in the study. Their baseline characteristics, including their age, sex, smoking history, SFCT, and αKl concentrations, are presented in Table 1. The mean ages of the participants and controls were similar, at 51.9 ±10.7 and 49.5 ± 10.4, respectively. Similarly, there was no significant difference in the sex distribution of the two groups. The prevalence of smoking

**Table 1. Baseline characteristics of the patients with CSC and controls.**

|  | CSC | Controls | $p$-value |
|---|---|---|---|
|  | (n = 56) | (n = 27) |  |
| Age (years) | 51.9±10.7 | 49.5±10.4 | 0.733 |
| Sex (Male/Female) | 46/10 | 19/8 | 0.223 |
| History of smoking, n (%) | 36 (43.4%) | 4 (4.82%) | 0.049 |
| αKl (pg/mL) | 827±232 | 724±183 | 0.035 |
| SFCT (μm) | 416±91.0 | 278±96.3 | <0.0001 |
| Acute/chronic CSC | 38/18 |  |  |

CSC, central serous chorioretinopathy; αKl, alpha-klotho; SFCT, subfoveal choroidal thickness. Data are mean ± SD or number (percentage) and were analyzed using Student's $t$-test or the Wilcoxon signed-rank test for continuous data and Fisher's exact test or the Pearson $\chi^2$ test for categorical data.

**Table 2. Comparison of the patients with acute or chronic CSC.**

| | Acute | Chronic | p-value |
| --- | --- | --- | --- |
| | (n = 38) | (n = 18) | |
| Age (years) | 50.1±10.2 | 55.7±11.0 | 0.231 |
| Sex (Male/Female) | 31/7 | 15/3 | 0.470 |
| History of smoking, n (%) | 24 (63.2%) | 12(66.7%) | 0.139 |
| αKl (pg/mL) | 877±226 | 721±214 | 0.010 |
| SFCT (μm) | 409±94.0 | 432±85.0 | 0.396 |
| POMS2 | 36.6±32.6 | 37.8±29.1 | 0.912 |

CSC, central serous chorioretinopathy; αKl, alpha-klotho; SFCT, subfoveal choroidal thickness; POMS, Profile of Mood States. Data are mean ± SD or number (percentage) and were analyzed using Student's *t*-test or the Wilcoxon signed-rank test for continuous data and Fisher's exact test or the Pearson $\chi^2$ test for categorical data.

was significantly higher in the CSC group than in the control group (43.4% and 4.82%, respectively; $p = 0.049$). The mean circulating concentration of αKl was significantly higher in the patients than in the control group (827 ± 232 and 724±183 pg/mL, $p = 0.035$). A propensity score-matched subpopulation was selected from the study sample by matching patients with CSC and controls according to smoking status. The αKl concentrations of patients with CSC and normal controls were compared, and the αKl concentration was found to be significantly higher in patients with CSC ($p = 0.033$). The mean SFCT was significantly greater in the CSC group than in the control group (416 ± 91.0 *vs.* 278 ± 96.3 μm, respectively; $p < 0.0001$). In the CSC group, 38 of the patients had acute CSC and 18 had chronic CSC.

Table 2 shows a comparison of the characteristics of the participants with acute or chronic CSC. There were no significant differences in the age, sex, smoking history, or SFCT of the two groups, with respective *p*-values of 0.120, 0.460, 0.646, and 0.175. However, the mean concentration of αKl was significantly higher in the acute CSC group than in the chronic CSC group (898±221 and 740 ± 224 pg/mL, $p = 0.038$). The mean POMS2 scores were similar (36.6 ± 32.6 for the acute CSC group and 37.8 ± 29.1 for the chronic CSC group, $p = 0.912$).

There were no relationships of αKl concentration with age, sex, smoking history, or SFCT in the CSC group, the control group, or the two CSC subgroups. However, a comparison of the αKl concentrations between patients with CSC and normal controls who were not smokers showed that they were significantly higher in the CSC group (888.9 ± 80.0 and 727.6 ± 92.0 pg/mL, $p = 0.017$). There were 10 (17.9%) cases of recurrence, including in the contralateral eye, of which five eyes were in the acute CSC group and five eyes were in the chronic CSC group. Of the 10 cases of recurrence, one eye had a previous history of CSC, and the patient showed spontaneous remission. Patients who experienced recurrent CSC had significantly higher αKl concentrations than those who did not ($p = 0.0219$). In the acute CSC group, the αKl concentration tended to be higher in patients that experienced recurrence (10 of 36 patients), although not significantly ($p = 0.0578$). In the chronic CSC group (20 patients), the αKl concentration was higher in patients that experienced recurrence (5 patients) than in those who did not (14 patients) ($p = 0.2340$) (Fig 1). In cases of recurrence, there were no significant differences in αKl by smoking habit status. Similarly, no significant difference in SFCT and αKl concentrations in patients with recurrent cases ($p = 0.999$).

## Discussion

In the present study, we found that the circulating concentrations of αKl are significantly higher in patients with CSC than in healthy adults, and significantly higher in those with acute

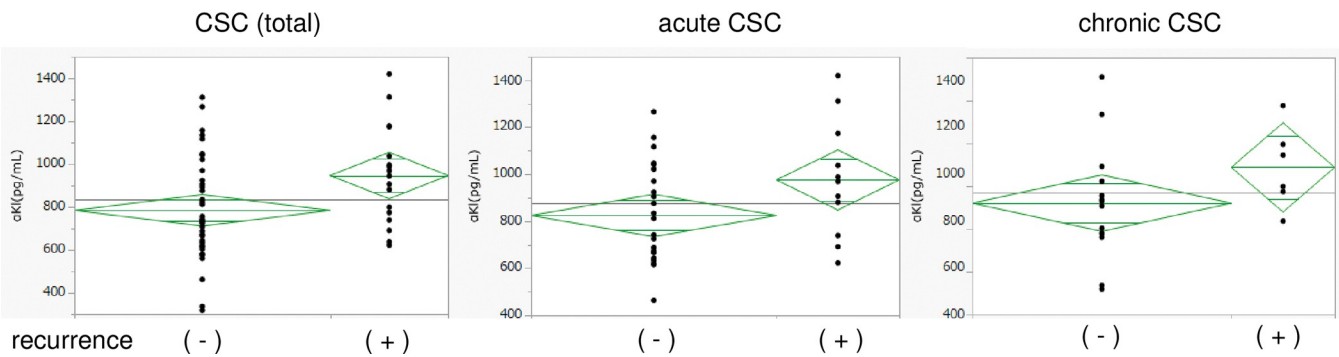

**Fig 1. Relationships between the presence or absence of recurrence and αKl concentration in the CSC, acute CSC, and chronic CSC groups.** The α-Klotho concentrations of the patients with CSC who experienced recurrence were significantly higher than in those who did not.

CSC than in those with chronic CSC. However, there was no significant difference in the concentrations of the chronic CSC subgroup and the control group. Studies have previously shown that the αKl concentration decreases with age and is lower in patients with diseases such as atherosclerosis and diabetes mellitus [9]. Blood pressure measurements were taken before FA/ICGA for CSC patients, we examined whether there was a difference in αKI according to blood pressure. There were 17 patients with a systolic blood pressure of ≥140 mmHg, who were defined as having hypertension. When these patients were compared with the patients with CSC who did not have hypertension, no significant difference in αKl concentration was identified ($p = 0.668$). In addition, the αKl concentration is high in patients with heart failure and decreases in response to treatment, suggesting that it may increase as part of the body's defense mechanisms [9]. The present findings suggest that αKl is not involved in the development of acute CSC, but that its concentration may increase as a result of this condition developing.

The underlying mechanism of CSC has been reported to involve high choroidal vascular permeability and choroidal vasodilation [10]. In the present study, there was no discernible relationship between αKI concentration and choroidal thickness, and this also suggests that the high αKI concentration may be a consequence, rather than a cause of, CSC, but the mechanism involved is unknown. Further research should be performed to determine whether there is a direct relationship or whether it is mediated through subretinal fluid infiltration. However, we found that the serum αKI concentrations of patients with acute CSC that experienced recurrence were higher than those of patients that did not. Because the αKI concentration is increased by stress [6], patients who experience recurrence are possibly under intermittent intense stress. A high αKI concentration is indicative of a maintained biological defense response, and may therefore be associated with a good prognosis, but given that the αKI concentrations of patients with acute CSC who experience recurrence are high, whether this finding is positive in such patients remains unclear. It may be that freedom from stress is associated with a better prognosis than the maintenance of defense mechanisms. It is possible that there is no relationship between this defense mechanism and recurrence, but further study is needed, including the measurement of αKI concentrations before and after the treatment of CSC.

The relationships of αKl concentration with age, sex, smoking history, and choroidal thickness were also evaluated in the CSC group, the control group, and in the acute CSC and chronic CSC groups, but no associations were identified. Although previous studies have shown a decrease in αKl with age [6, 11], we did not identify this trend in the present study. This may be because the participants in the two groups were matched for age, such that there

was no significant difference in the age distributions of the two groups. We also found no association between αKl concentration and smoking. A previous study has shown that the αKl concentrations of older smokers tend to be higher than those of older non-smokers (60.3 ± 1.7 years of age), whereas no difference was identified for middle-aged individuals (46.1 ± 5.1 years of age) [11]. In the present study, only middle-aged people were included, which may account for the absence of a significant relationship between αKl concentration and smoking.

Nakanishi *et al*. have suggested that an increase in αKI with age is paradoxical, because it is considered to be anti-aging molecule, and instead that this increase may be a compensatory response to stress, and that αKl may be an anti-inflammatory molecule. Previous studies have shown that psychological stress can result in increases in the concentrations of proinflammatory molecules, such as interleukin (IL)-6 [12], and the concentrations of αKl and IL-6 have been shown to positively correlate [11]. The patients in the present study are likely to have been affected by inflammation.

There was no difference in the POMS scores of the participants with acute or chronic CSC, nor between the CSC group and the control group. With respect to the components of the POMS questionnaire, the CSC group exhibited high values for VA, indicating vigor; vitality; and F, indicating friendliness; and low values for AH, indicating anger; DD, indicating depression; FI, indicating fatigue; and apathy. This suggests that negative emotions are not directly related to the development of CSC. Although not statistically significant, the Fatigue and Apathy scores tended to be higher in the acute CSC group than in the chronic CSC group, which may imply that fatigue is more marked early in the course of CSC. We found no significant relationship between αKl concentration and the POMS score, but because both variables are influenced by a multitude of factors, this warrants further investigation.

The present study was limited by its retrospective nature, the fact that it was conducted at a single institution, and its relatively small size. We had aimed to collect 50 samples from both CSC and controls; however, after matching and the application of the exclusion criteria, the number of controls available for inclusion was lower. If the sample size were to be increased, the significance levels associated with the results would likely change. Furthermore, the classification of acute and chronic CSC was based on the self-reported onset of symptoms, which may have differed from the reality.

In conclusion, we have shown that the circulating αKl concentrations of patients with CSC are high.

## Supporting information

**S1 Data.**
(XLSX)

## Acknowledgments

We thank Mark Cleasby, PhD, from Edanz (https://jp.edanz.com/ac) for editing drafts of this manuscript.

## Author Contributions

**Conceptualization:** Yuki Yamamoto.

**Data curation:** Eri Tahara, Yuki Yamamoto, Takaaki Sugisawa.

**Formal analysis:** Eri Tahara, Yuki Yamamoto.

**Investigation:** Fumi Gomi.

**Writing – original draft:** Yuki Yamamoto.

**Writing – review & editing:** Fumi Gomi.

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
