## [Decision Letter · Decision Letter 0]

3 Mar 2023

PONE-D-23-04718Patients with central serous chorioretinopathy have high circulating alpha-klotho concentrationsPLOS ONE

Dear Dr. Yamamoto,

Thank you for submitting your manuscript to PLOS ONE. After careful consideration, we feel that it has merit but does not fully meet PLOS ONE’s publication criteria as it currently stands. Therefore, we invite you to submit a revised version of the manuscript that addresses the points raised during the review process.

We look forward to receiving your revised manuscript.

Kind regards,

Tatsuya Inoue

Academic Editor

PLOS ONE

Journal Requirements:

Please include your amended statements within your cover letter; we will change the online submission form on your behalf."

5. Please upload a new copy of Figure 1 as the detail is not clear. Please follow the link for more information: https://blogs.plos.org/plos/2019/06/looking-good-tips-for-creating-your-plos-figures-graphics/" https://blogs.plos.org/plos/2019/06/looking-good-tips-for-creating-your-plos-figures-graphics/

Additional Editor Comments:

Thank you for submitting the manuscript to PLOS ONE. This study is interesting and challenging although the molecular mechanism of alpha-klotho still remains unclear. For the final decision, the authors should address the issues raised by the reviewers.

As the reviewers mentioned, there is a possibility that some background factors (age, smoking, acute/chronic CSC etc.) influenced the values of alpha-klotho. Therefore, in order to clarify the importance of measuring alpha-klotho, the authors should perform multivariate analysis for CSC versus control groups using logistic regression analysis.

Reviewers' comments:

Reviewer's Responses to Questions

**Comments to the Author**

1. Is the manuscript technically sound, and do the data support the conclusions?

Reviewer #1: No

Reviewer #2: Yes

2. Has the statistical analysis been performed appropriately and rigorously? 

Reviewer #1: No

Reviewer #2: Yes

3. Have the authors made all data underlying the findings in their manuscript fully available?

Reviewer #1: No

Reviewer #2: Yes

4. Is the manuscript presented in an intelligible fashion and written in standard English?

Reviewer #1: No

Reviewer #2: Yes

5. Review Comments to the Author

Reviewer #1: The authors reported that the concentration of alpha-klotho (αKl) in circulating plasma is significantly higher in patients with CSC than in controls, and thatαKl was evaluated as a potential marker for CSC. The analysis is important to understand of the pathogenesis of CSC. However, there seem to be several serious problems with interpretation.

#1. There is insufficient information provided regarding the study design. For example, a more detailed description of inclusion and exclusion criteria, the number of patients excluded, and how subjects were selected is needed. How was randomization performed? There is also no mention of the timing of blood sampling or whether treatment was performed for acute or chronic CSC, among other things

#2. In this study, there is a significant difference in smoking history as a background factor. Smoking is a known risk factor for CSC. Therefore, the author should be matched as a background factor.

#3. The recurrence of CSC cases was also evaluated in this study, and it is necessary to describe whether there is a history of treatment for CSC and the treatment methods used.

#4. The author should evaluate the relation between smoking and αKl in recurrence cases.

#5. The authors may also need a larger sample size. If the sample size increased, the statistical results will be different and became not significant.

#6. The author described that no significant relationship was found between αKI concentration and SFCT in the analysis of all cases of CSC, but what about in the case of recurrent CSC?

Reviewer #2: This manuscript by Tanaka and colleagues provides a potential insight to consider the association of circulating alpha-klotho with central serous chorioretinopathy (CSC), both of which are supposed to be related to psychological stress. Although their cause-effect relationship still remains unknown, the data presented here are entirely novel and so important in terms of the possibility of stimulating future studies in this field. The authors are invited to respond to a few minor points as follows:

1. It would be interesting to compare CSC patients with no history of smoking (n=20) and control subjects with no history of smoking (n=23), because alpha-klotho is reported to reflect smoking status.

2. Is there any correlation between alpha-klotho and other biomarkers in the systemic circulation? There are several systemic diseases reported to affect alpha-klotho, most of which also influence, or are influenced by, systemic biochemical markers.

3. There are some reports stating that hypertension is related to CSC. Please show blood pressure or a history of hypertension in this case series.

6. PLOS authors have the option to publish the peer review history of their article (what does this mean?). If published, this will include your full peer review and any attached files.

Reviewer #1: No

Reviewer #2: No

---

## [Author Response · Author response to Decision Letter 0]

6 Aug 2023

Dear Dr. Chenette,

Thank you for taking the time to review this manuscript. We appreciate the prompt review and the opportunity to revise and improve the manuscript further. We are grateful to the two reviewers for the time and effort that they have spent reviewing the manuscript. Please find our point-by-point responses to the comments below. The text shown in red below is a copy of the modified parts of the manuscript, and these are shown as tracked changes in the revised version, which is attached.

Responses to Reviewer #1

#1. There is insufficient information provided regarding the study design. For example, a more detailed description of inclusion and exclusion criteria, the number of patients excluded, and how subjects were selected is needed. How was randomization performed? There is also no mention of the timing of blood sampling or whether treatment was performed for acute or chronic CSC, among other things

Response: We apologize for not fully describing the study design. We thank the reviewer for raising this issue. The study was retrospective in nature, as described on line 67 and randomization was not performed.

With respect to the inclusion and exclusion criteria, we have added the following explanation (Lines 74–80, page 5):

“We included patients with both acute and chronic CSC, which we categorized based on a duration of symptoms of less or more than 6 months, respectively. Patients with macular neovascularization detected using optical coherence tomography angiography (OCTA) at baseline; those who had used steroids; those with systemic diseases, such as renal or cardiac disorders, which affect αKl concentration; those with a history of anti-vascular endothelial growth factor drug therapy; and those with severe myopia of >−6.0 D were excluded.”

Five patients with macular neovascularization, two with steroid use, two with systemic disease, and three with strong myopia were excluded.

We have added the following regarding the timing of the blood sampling:

“To ensure uniformity of the conditions, after a diagnosis of CSC had been made by FA and ICGA, blood samples were drawn approximately 6 hours after waking from patients who gave their consent, and at the time consent was given in controls” (Lines 87–92, page 5-6).

We have also added the initial treatments administered to lines 101–104 page 6.

#2. In this study, there is a significant difference in smoking history as a background factor. Smoking is a known risk factor for CSC. Therefore, the author should be matched as a background factor.

Response: We apologize for not matching the patients with regard to their smoking history. To account for potential confounding by the association between smoking and CSC, variables potentially associated with smoking, including sex and age, were included in a logistic regression model in which smoking was the dependent variable, to calculate a propensity score for CSC. A propensity score-matched subpopulation was then selected from the study sample by matching patients with CSC and controls who were smokers. The αKI concentrations were then compared between these two groups, and the αKI concentration was found to be significantly higher in the patients with CSC (p=0.033).

We have added the following explanation to lines 134–138, page 7:

“A propensity score-matched subpopulation was selected from the study sample by matching patients with CSC and controls according to smoking status. The αKI concentrations of patients with CSC and normal controls were compared, and the αKI concentration was found to be significantly higher in patients with CSC (p=0.033).”

#3. The recurrence of CSC cases was also evaluated in this study, and it is necessary to describe whether there is a history of treatment for CSC and the treatment methods used.

Response: We thank the reviewer for their recommendation. Of the 10 cases of recurrence, one eye had a previous history of CSC, and the patient showed spontaneous remission. However, we did not have records of any previous history of CSC, but instead received self-reports at the patient’s first visit to our clinic. Therefore, this information may not have been very accurate.

We have added the following explanation to lines 168–169, page 9:

Of the 10 cases of recurrence, one eye had a previous history of CSC, and the patient showed spontaneous remission.

#4. The author should evaluate the relation between smoking and αKl in recurrence cases.

Response: We thank the reviewer for this suggestion. In cases of recurrence, three of the 10 eyes were in patients with a smoking habit. In cases of recurrence, there were no significant differences in αKI by smoking habit status.

We have added an explanation to the manuscript as follows (lines 174–175, page 10):

“In cases of recurrence, there were no significant differences in αKI by smoking habit status..”

#5. The authors may also need a larger sample size. If the sample size increased, the statistical results will be different and became not significant.

Response: We agree that the relatively small sample size was the major limitation of the study. We had aimed to collect 50 samples from both CSC and controls; however, after matching the participants and applying the exclusion criteria, the number was lower in the normal group. In response to this comment, we have strengthened our discussion of the limitations of the study.

We have added the following explanation (Lines 247–250, page 13):

“We had aimed to collect 50 samples from both CSC and controls; however, after matching and the application of the exclusion criteria, the number of controls available for inclusion was lower. If the sample size were to be increased, the significance levels associated with the results would likely change.”

#6. The author described that no significant relationship was found between αKI concentration and SFCT in the analysis of all cases of CSC, but what about in the case of recurrent CSC?

Response: We thank the reviewer for this constructive remark. There was no significant relationship between αKl and SFCT in participants who experienced recurrence (p=0.999).

 We have added the following explanation (Lines 175–176, page 10):

“Similarly, no significant difference in αKI concentrations and SFCT in patients with recurrent cases. (p=0.999)”

We thank the reviewer for the comments.

 

Responses to Reviewer #2

1. It would be interesting to compare CSC patients with no history of smoking (n=20) and control subjects with no history of smoking (n=23), because alpha-klotho is reported to reflect smoking status.

Response: We thank the reviewer for carefully reading the manuscript and the suggestion. We have performed an analysis of patients who were and were not smokers in the CSC and control groups, and the αKl concentration was found to be significantly higher in the CSC group (888.9 ± 80 and 727.6 ± 92pg/mL, respectively; p = 0.017).

We have added this finding to lines 163–166, page 9, as follows:

“However, a comparison of the αKl concentrations between patients with CSC and normal controls who were not smokers showed that they were significantly higher in the CSC group (888.9 ± 80.0 and 727.6 ± 92.0 pg/mL, p = 0.017).

2. Is there any correlation between alpha-klotho and other biomarkers in the systemic circulation? There are several systemic diseases reported to affect alpha-klotho, most of which also influence, or are influenced by, systemic biochemical markers.

Response: We thank the reviewer for this question.

We apologize that we only checked the status of both patients with CSC and normal controls with respect to the presence of underlying disease, such as heart failure and kidney dysfunction, and the use of oral medication in the present study. In addition, we recorded each patient’s medical history but did not evaluate heart failure or renal function using laboratory investigations. Previous studies have shown that αKI is associated with both renal and heart failure. As the reviewer points out, we could have made interesting findings if we had measured parameters related to renal function and blood pressure, for example.

We comment regarding hypertension below.

3. There are some reports stating that hypertension is related to CSC. Please show blood pressure or a history of hypertension in this case series.

Response: We thank the reviewer for this constructive remark.

Blood pressure was measured in all of the patients at the time of the FA/ICGA, and there were 17 patients with a systolic blood pressure of ≥140 mmHg, who were defined as having hypertension. When these patients were compared with the patients with CSC who did not have hypertension, no significant difference in αKl concentration was identified (p=0.668).

We have added this finding to lines 189–194, page 10, as follows:

“Blood pressure measurements were taken before FA/ICGA for CSC patients, we examined whether there was a difference in αKI according to blood pressure. There were 17 patients with a systolic blood pressure of ≥140 mmHg, who were defined as having hypertension. When these patients were compared with the patients with CSC who did not have hypertension, no significant difference in αKl concentration was identified (p=0.668)

We thank the reviewer for the comments.

We have attempted to improve the manuscript in accordance with the reviewers’ comments. We sincerely appreciate their constructive comments and hope that the revised manuscript will now be acceptable for publication in PLOS ONE. We would like to thank you for permitting us to submit a revised version of the manuscript.

Yours sincerely,

Yuki Yamamoto

---

## [Decision Letter · Decision Letter 1]

30 Nov 2023

Patients with central serous chorioretinopathy have high circulating alpha-klotho concentrations

PONE-D-23-04718R1

Dear Dr. Yamamoto,

We’re pleased to inform you that your manuscript has been judged scientifically suitable for publication and will be formally accepted for publication once it meets all outstanding technical requirements.

Kind regards,

Tatsuya Inoue

Academic Editor

PLOS ONE

Additional Editor Comments (optional):

Reviewers' comments:

Reviewer's Responses to Questions

**Comments to the Author**

1. If the authors have adequately addressed your comments raised in a previous round of review and you feel that this manuscript is now acceptable for publication, you may indicate that here to bypass the “Comments to the Author” section, enter your conflict of interest statement in the “Confidential to Editor” section, and submit your "Accept" recommendation.

Reviewer #1: All comments have been addressed

2. Is the manuscript technically sound, and do the data support the conclusions?

Reviewer #1: Partly

3. Has the statistical analysis been performed appropriately and rigorously? 

Reviewer #1: Yes

4. Have the authors made all data underlying the findings in their manuscript fully available?

Reviewer #1: Yes

5. Is the manuscript presented in an intelligible fashion and written in standard English?

Reviewer #1: Yes

6. Review Comments to the Author

Reviewer #1: (No Response)

7. PLOS authors have the option to publish the peer review history of their article (what does this mean?). If published, this will include your full peer review and any attached files.

Reviewer #1: No

---

## [Editor Report · Acceptance letter]

13 Dec 2023

PONE-D-23-04718R1 

PLOS ONE

Dear Dr. Yamamoto, 

I'm pleased to inform you that your manuscript has been deemed suitable for publication in PLOS ONE. Congratulations! Your manuscript is now being handed over to our production team.

Kind regards, 

on behalf of

Dr. Tatsuya Inoue 

Academic Editor

PLOS ONE